# OPEN-DOMAIN TEXT EVALUATION VIA CONTRASTIVE DISTRIBUTION METHODS

## ABSTRACT

Recent advancements in open-domain text generation, driven by the power of large pre-trained language models (LLMs), have demonstrated remarkable performance. However, assessing these models' generation quality for specific attributes remains a challenge. Traditional reference-based metrics like BLEU, ROUGE, and METEOR measure the similarity between machine-generated outputs and human-written references, which deviates from the principle of open-ended generation tasks, leading to low correlation with human judgments. While trainable discriminator-based evaluation metrics show promise, the acquisition of high-quality training data presents a formidable obstacle. In this paper, we introduce a novel method for evaluating open-domain text generation called Contrastive Distribution Methods (CDM). Leveraging the connection between increasing model parameters and enhanced LLM performance, CDM creates a mapping from the *contrast* of two probabilistic distributions – one known to be superior to the other – to quality measures. We investigate CDM for open-domain text generation evaluation under two paradigms: 1) *Generative* CDM, which harnesses the contrast of two language models' distributions to generate synthetic examples for training discriminator-based metrics; 2) *Discriminative* CDM, which directly uses distribution disparities between two language models for evaluation. Our experiments on multi-turn dialogue and factuality in abstractive summarization demonstrate that CDM correlate better with human judgment than existing automatic evaluation metrics on both tasks, highlighting the strong performance and generalizability of our approach.

## 1 INTRODUCTION

In recent years, open-domain text generation, fueled by large pretrained generative language models (LLMs), has made significant advancements, garnering substantial attention (Radford et al., 2018; 2019; Brown et al., 2020; OpenAI, 2022; 2023). These systems have showcased remarkable capabilities, such as producing human-like responses, contributing to natural language comprehension, and even performing complex tasks like programming and content generation. With the empirical success, the development of reliable and scalable automatic evaluation metrics for these models become imperative, yet the problem remains an unresolved challenge.

Existing automatic evaluate metrics from pre-LLM eras have their respective limitations. Specifically, reference-based statistical metrics (e.g. BLEU (Papineni et al., 2002), ROUGE (Lin, 2004), METEOR (Banerjee & Lavie, 2005)) do not work well for open-ended generation problems with high content diversity like storytelling (Yao et al., 2019) and dialogue systems (Mesgar et al., 2019; Li et al., 2017; Wen et al., 2016), as for these tasks, it is challenging, if not impossible, to collect a sufficiently large number of reference examples to represent the distribution of all feasible outputs. Therefore, prior works have shown their low correlation with human judgments (Liu et al., 2016; Hu et al., 2020). With recent progress in pretrained models, model-based reference metrics like BERTScore (Zhang et al., 2019), Bluert (Sellam et al., 2020) are proposed to facilitate automatic evaluation for text generation. They alleviate the sample efficiency issue of statistical reference-based methods by using pretrained models to compute the similarities between texts based on higher-level semantics. However, the effectiveness of such methods is still reliant on the representativeness of the reference set, and thus falls short when the output semantic space is also highly diverse.

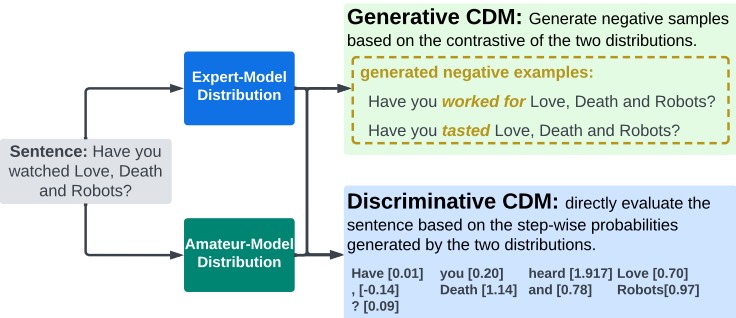

Figure 1: Conceptual illustration of the Contrastive Distribution Methods (CDM). (a) *Generative CDM* generates negative examples for training a discriminator-based metric. (b) *Discriminative CDM* directly evaluate the distribution/sequence in a Naive Bayes framework.

Reference-free evaluation metrics, which assess text directly and provide a score, offer a more reliable solution for automatically evaluating open-domain text generation. There are generally two paradigms for training models to evaluate text without references: 1) *Discriminator-based approaches*, as seen in methods like ADEM (Lowe et al., 2017) and DEAM (Ghazarian et al., 2022), treat the problem as a prediction task. These approaches train a classifier or regressor to generate a score as the quality assess. However, these methods typically require extensive human annotations or involve dedicated manual designs for generating negative samples to train the classifier. 2) *Distribution/Divergence-based approaches* (Pillutla et al., 2021; Pimentel et al., 2022) focus on obtaining a continuous divergence score between distributions. This approach has shown promising results in system-level evaluations. However, these methods often face challenges when it comes to accurately assigning credit to individual data points, limiting their ability to perform instance-level evaluations. There are also several existing model-based metrics (Zhong et al., 2022a; Liu et al., 2023b) built from modules under both paradigms. However, they still suffer from the inherent limitations of these two paradigms.

In this paper, we propose Contrastive Distribution Methods (CDM), a general and reference-free framework for evaluating open-domain text generation. CDM operates on a straightforward yet broadly applicable premise: models with similar architectures but varying sizes generally exhibit improved generation quality as model size increases. Consequently, CDM is designed to capture the dynamics of model performance as it scales with the increasing number of parameters. Utilizing such dynamics, CDM *contrasts* two language models' distributions and conduct inference in both generative and discriminative manners, enabling the creation of automatic evaluation metrics.

CDM improves over previous reference-free evaluation metrics in two aspects: 1) The generative CDM as illustrated in the upper right corner of Figure 1 produces effective negative samples to facilitate the learning of discriminator-based evaluation metrics without the requirement of additional human annotations or sophisticated design for the data generation process. 2) The discriminative CDM as illustrated in the lower right corner of Figure 1 provides a distribution-level measurement without sacrificing the capability to assign credits to individual sentences, and thus results in more reliable instance-level evaluation metrics. Results on dialogue coherence and abstractive summarization evaluation show that, CDM achieves strong performance and outperform existing methods on all datasets without much task-specific designs.

## 2 BACKGROUND AND RELATED WORKS

**Open-Domain Text Evaluation** There has been a synchronously growing interest in developing robust evaluation methods for open-domain text generation models. Traditional evaluation metrics, such as BLEU and ROUGE, have been shown to be inadequate for assessing the quality of complex, multi-sentence responses generated by these models. As a result, researchers have explored alternative evaluation methods, including human evaluation, adversarial evaluation, and unsupervised metrics. Human evaluation remains the gold standard, but it is time-consuming and costly. Adversarial evaluation, which involves testing models against a set of challenging examples, has shown promise in

identifying weaknesses in current models. Unsupervised metrics, such as BERTScore and Perplexity, provide quick and automated evaluation, but their correlation with human judgments remains a topic of debate. The field of open-domain text evaluation continues to evolve, and developing reliable evaluation methods will be essential for advancing the state-of-the-art in this exciting area of research.

**Discriminator-based Metrics** ADEM (Lowe et al., 2017) is one of the first attempts at training a model to evaluate machine-generated text. It deals with single-turn dialogue evaluation problem, and uses the contextualized representation of the context in interaction with that of the responses to train the model. DEAM (Ghazarian et al., 2022) is a novel evaluation metric that aims to assess the coherence and quality of multi-turn dialogue systems. Unlike traditional evaluation metrics, DEAM uses manipulation techniques to construct negative samples from positive samples, allowing for a more nuanced assessment of model performance. DEAM operates by first parsing the sequence into an abstract meaning representation (AMR), and then manipulating the AMR to introduce inconsistencies and irrelevancies that undermine the coherence of the dialogue. The manipulated AMR is then transformed back into text form for evaluation. This method supports multi-turn dialogue evaluation and has achieved state-of-the-art performance on various benchmark datasets. By using AMR-based semantic manipulations, DEAM provides a promising approach for evaluating the quality of dialogue systems in a more comprehensive and accurate manner. *Generative* CDM shares a similar process , as it manipulates the positive true samples for the generation of negative samples, serving the purpose of training a classifier.

**Distribution/Divergence-based Metrics** MAUVE and follow-up works (Pillutla et al., 2021; Pimentel et al., 2022) analyse the quality gap between human-generated text and machine-generated text by studying the divergence frontier of human-generated samples in contrast to the learnt model. While their setup is not directly relevant to our approach, it provides an insightful perspective of using the likelihood predictions of LMs for evaluation purposes. Zhong et al. (2022a) proposes a multi-dimensional evaluation system for more robust automatic evaluation. It ensembles the score from a set of *discriminator-based* metrics, each of which trained to evaluate a specific aspect in intuition of the text quality. GPTEval (Liu et al., 2023b) tries to quantitatively exploit large language models that are trained with strong human alignment. It uses the score prediction from GPT-4 (OpenAI, 2023) to evaluate how well the given text adheres to human opinion. *Discriminative* CDM falls under this paradigm, since it serves as a metric with more continuously distributed scores for the evaluated text.

**Contrastive Decoding and Contrastive Momentum** Contrastive decoding is a decoding algorithm that leverages the strengths of two language models: a stronger expert model and a weaker amateur model. The algorithm decodes towards the objective of maximizing the difference between the log-probabilities of the expert and amateur models, resulting in high-quality generated samples. Specifically, the algorithm tries to decode sequences that maximize the *contrastive momentum*:

$$\log p_{\mathrm{e}}(x) - \log p_{\mathrm{a}}(x), \tag{1}$$

where $p_{\mathrm{e}}$ and $p_{\mathrm{a}}$ represent the expert and the amateur models, respectively, and $x$ is the generated sample. The original paper (Li et al., 2022) demonstrates that this approach results in higher quality samples than decoding from the expert model alone. Contrastive decoding provides an insightful way to study the dynamics of how models' capabilities scale up with larger parameter numbers. The proposed CDM is highly inspired by the Contrastive decoding method, yet leveraging it for evaluation purposes.

## 3 METHODOLOGY

### 3.1 NOTATIONS AND PROBLEM FORMULATION

We use $\mathbf{s}$ to denote a sequence and $s_i$ to denote the $i-$th token in $\mathbf{s}$. $p(\mathbf{s})$ denotes the probability of sequence $\mathbf{s}$ under a model $p$. We assume model $p$ is a probabilistic distribution defined on $\Sigma^*$, where $\Sigma$ is the set of valid tokens and $\Sigma^*$ is the universal set of all sequences consisting of such tokens.

Consider an imaginary distribution-level oracle metric $E(p)$ which projects from a model distribution $p(\mathbf{s})$ to "a measure of model performance" – a scalar. This function does not necessarily have an analytical form, however, we assume that we have access to some partial order relations it defines. Intuitively, this imaginary oracle $E(p)$ should correlate perfectly with human judgements of the

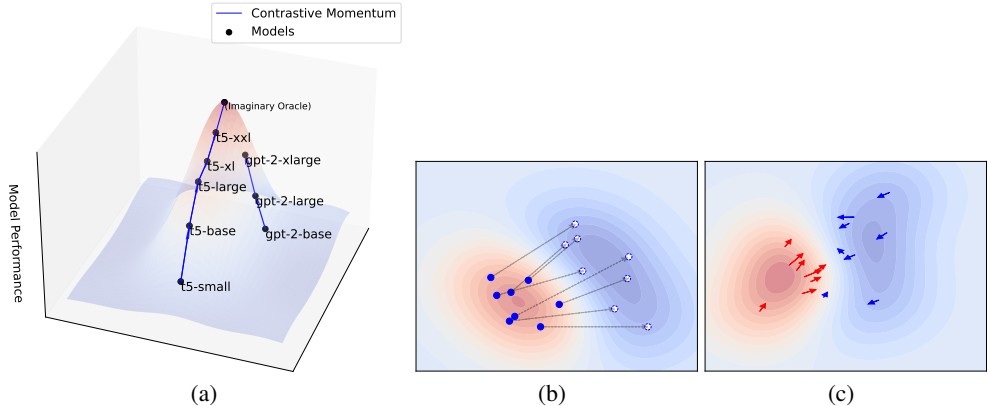

Figure 2: (a) While it is hard to assume a total order for models from different model classes under the oracle metric $E(p)$, it is plausible to assume partial orders for models from the same model class. (b) Generative CDM uses the degraded distribution $p_a^-$ to synthesize fake samples for training a discriminator as the metric. The warm/cold region indicates the decision boundary of the resulting trainable metric induced by fake samples from $p_a^-$. (c) Discriminative CDM directly determines the decision boundary by pooling the values of the step-wise contrastive momentum.

model performance, and any evaluation metric that correlates better with human judgments is a better approximation of $E(p)$.

With the notion of oracle $E(p)$, we can perform:

- *Discriminative* inference:

  a) **(Distribution-level evaluation)** to evaluate any existing models by ranking them according to $E(p)$

  b) **(Sample-level evaluation)** to quantify the contribution of a specific sample $\mathbf{s} \sim p(\mathbf{s})$ *w.r.t.* $E(p)$

- *Generative* inference: to improve or degenerate the generation quality by altering $p$ towards maximization or minimization of $E(p)$. The generated examples can then be used to train discriminator-based sample-level evaluation metrics.

In this paper, we explore both discriminative and generative inference of CDM for automatically evaluation of open-domain generation.

## 3.2 THE PARTIAL ORDER ASSUMPTION

We hereby discuss in detail how we can conduct contrastive methods for evaluation purposes. While it is nontrivial to come up with analytical forms for $E(p)$, we can make some assumptions to obtain partial orders from $E(p)$. Consider a *series* of models that share similar architectures and other pretraining/finetuning setups, but differ in model sizes (e.g. T5-small/base/large, etc.). It is usually safe to assume that the model with a larger number of parameters perform better than the smaller one under most aspects. More formally, we can assume a partial order (a linear order within one concerned model class) induced by the oracle metric $E(p)$ as illustrated in Equation 2 and Figure 2(a):

$$E(p_{small}) < E(p_{base}) < E(p_{large}) \tag{2}$$

## 3.3 FIRST ORDER APPROXIMATION OF $E(p)$

As is previously mentioned in Sec 3.1, it could be intractable to compute $\frac{\partial E(p)}{\partial \log p}$ since we do not assume knowing the analytical form of $E(p)$. Following similar approach as in Li et al. (2022), we approximate it using a secant hyperplane between the *amateur* distribution $p_a$ and the *expert* distribution $p_e$. In other words, we assume $E(p)$ follows the following analytic form:

$$E(p) \propto \sum_{\mathbf{s}} \Big( \log p_e(\mathbf{s}) - \log p_a(\mathbf{s}) \Big) \log p(\mathbf{s}) \ \ s.t. \ \ E(p_e) > E(p_a), \tag{3}$$

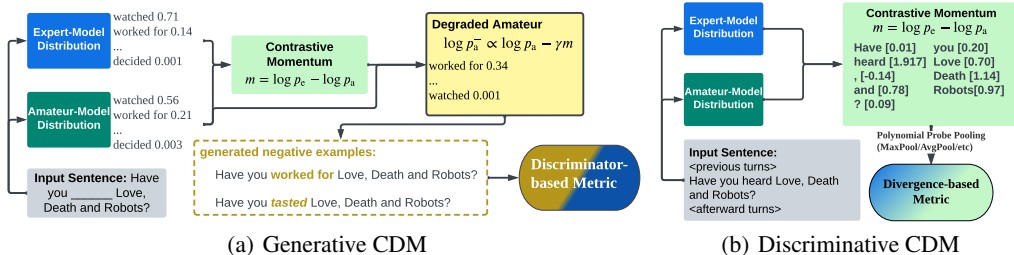

Figure 3: A more detailed illustration of the two Contrastive Distribution Methods (CDM). (a) *Generative CDM* constructs fake negative samples from positive ones for training a discriminator-based metric. (b) *Discriminative CDM* directly evaluate the distribution/sequence in a Naive Bayes framework.

We can further define $m(\mathbf{s}) = \log p_e(\mathbf{s}) - \log p_a(\mathbf{s})$ as the contrastive momentum. Different choices of $p_a$ and $p_e$ result in different quality of the first-order approximations for $E(p)$, hence different performance of the evaluation metric. We need to investigate the general principle for choosing the expert and amateur distributions.

## 3.4 CONTRASTIVE DISTRIBUTION METHODS

### 3.4.1 GENERATIVE CDM: SYNTHETIC DATA GENERATION WITH CDM

Generative CDM follows prior works such as ADEM (Lowe et al., 2017) and DEAM (Ghazarian et al., 2022) to formulate reference-free evaluation metrics as prediction tasks. In order to evaluate generated texts, a discriminator can be trained on positive and negative examples to serve as the evaluation metric.[1] However, while we can assume human-written texts are positive examples, negative examples are non-trivial to obtain. Randomly generating negative examples using a uniform distribution over all possible sequences of tokens is not efficient, as most negative examples generated this way would be too trivial. On the other hand, generating negative examples from pretrained large-language models may not result in real low-quality texts, which would confuse the discriminator.

Generative CDM provides a controllable approach to reduce the quality of existing pretrained language models to generate "sufficiently deceptive negative examples". Following our mathematical formulation, this is effectively equivalent to descending along the direction of $-\frac{\partial E(p)}{\partial \log p}$ from the weaker amateur model $p_a$. We can utilize the previous approximation which is to replace the differential operation with the difference between $p_a$ and a selected expert distribution $p_e$. Specifically, by leveraging the contrastive momentum $m = \log p_e - \log p_a$ in the reversed direction, we controllably degenerate from the *amateur* model. Consequently, we obtain a probability distribution $\log p_a^- \propto \log p_a - \gamma m$ that disproportionately amplifies the likelihood of "machine artifacts" in a controllable (by setting $\gamma$) scale. Sampling from $p_a^-$ allows us to obtain suitable negative examples.

We hereby discuss how to generate targeted negative examples. The whole process can be viewed as a controllable generation problem. We start from existing positive examples $\mathbf{s}$ and selectively decrease the overall text quality without altering the majority of the context and meaning. As a result, the generated negative examples would be more disguising compared to sampling from $p_a^-$ at the very beginning. To achieve this, we train a *segment insertion/reconstruction* model. Given a positive example and the position at which a segment is removed (randomly or strategically), we model a conditional distribution that reconstructs the original segment. Once we have obtained such a model and its degenerated form, we can utilize it to repeatedly operate on existing positive examples, allowing us to construct negative examples in a further well-controlled manner. Figure 3(a) and 2(b) illustrates the process. This enables us to generate negative examples that are specifically targeted and deceptive.

---

[1]The discriminator does not necessarily need to provide binary decisions, it can also produce scores. But we use binary examples for simplicity.

### 3.4.2 DISCRIMINATIVE CDM: DIRECTLY USING CDM AS THE METRIC

Although Generative CDM is a reasonably flexible and scalable framework, there are still many variable factors in the generation process (e.g. how to choose which segment to remove, the randomness in the process of segment reconstruction, degradation strength factor $\gamma$ etc.) that may affect the performance of the resulting metric. Therefore, it is desirable to remove the generation subroutine completely. Consider why we need to train a discriminator as a metric, because we usually do not have a tractable model for the positive or negative distribution. However, under the CDM framework, we do have a tractable model $p_a^-$ for the negative distribution, which is composed from the amateur model $p_a$ and the contrastive momentum $m$. In light of this, we can consider directly deploying $m$ as a divergence-based metric for evaluation.

For each sequence, we concern the step-wise contrastive momentum $m(x_t|\mathbf{s}_{<t}) = \log p_e(x|\mathbf{s}_{<t}) - \log p_a(x|\mathbf{s}_{<t})$ composed from the *amateur* model and the *expert* model. A good sample, for which both models' likelihood prediction will be relatively high, $\sum_t \log m(x|\mathbf{s}_{<t})$ should be significantly larger than 0 in the ideal case. If we strictly follow the definition, summing up the step-wise contrastive momentum over the entire sequence (*i.e.* sum-pooling) would be the metric to evaluate the generation quality. See Figure 3(b) and 2(c).

However, we argue that there would be a subtle discrepancy between theory and practice. First, the sum-pooled score would be numerically influenced by the sequence length (even though not as severely as directly using the log-likelihood from either model). Moreover, sum-pooling overemphasizes the impact of extremely low probability steps because the discrepancy between the amateur model and expert model predictions in low-probability regions could be significantly amplified on the logarithmic scale.

In the experiments, we compare different strategies to *pool* the sequence of step-wise contrastive momentum values into a sentence-level evaluation score. We call this paradigm of using CDM as Discriminative CDM. There is no explicit generation process in Discriminative CDM as we only treat the two models and their contrastive momentum as likelihood value predictors.

## 4 EXPERIMENTS

### 4.1 DIALOGUE EVALUATION WITH CDM

The first part of our experiment is primarily focused on dialogue evaluation. Given a set of annotated dialogues, each with human-annotated quality scores ranging from $0.0$ to $1.0$, our objective is to assign scores to each evaluated sequence that maximizes the correlation with human annotations. Additionally, for dialogue evaluation, we assume we are not permitted to perform any training on data within the same domain. Our training/fine-tuning exercises are conducted on dialogues from both TopicalChat (Gopalakrishnan et al., 2019) and PersonalChat datasets (Zhang et al., 2018). Subsequently, we evaluate our methods on annotated dialogues from the FED (Mehri & Eskenazi, 2020) and DSTC9 (Gunasekara et al., 2020) datasets.

**Dataset and Experiment Setup** We adopt most experimental settings from DEAM (Ghazarian et al., 2022) to verify the effectiveness of our method. The statistics of the involved datasets in our experiments are shown as follows:

Table 1: Data usage in our experiments

| **Dataset** | **size** | **Avg. len** |
| --- | --- | --- |
| TopicalChat + PersonalChat (Gopalakrishnan et al., 2019; Zhang et al., 2018) | 17567/2078 | 377 |
| FED (test, w/ human annotation) (Mehri & Eskenazi, 2020) | 125 | 168 |
| DSTC9 (test, w/ human annotation) (Gunasekara et al., 2020) | 2200 | 318 |

The two models in each contrastive and expert model from different sizes of T5 (Raffel et al., 2019; Wei et al., 2021) checkpoints respectively. For further study of how different choices of such model pairs and other factors impact the performance of CDM, please refer to Table 3. Note that the smallest configurations (typically 7b) of state-of-the-art LLMs, such as LLaMa-1/2(Touvron et al., 2023a;b), can still be too strong to serve as the amateur model in CDM. Further more, such LLMs are usually

causal/auto-regressive ones, lacking the ability to more flexibly manipulate the text/calculate the likelihood of a subsequence. We look forward to new LLMs with a longer range of published scales.

### 4.1.1 MODEL SPECIFICATION

There would be multiple strategies to construct the context-prediction pairs. We study the following cases for Generative CDM:

- Segment-Single: The manipulation of data is only applied once to a random segment no longer than 20 tokens in a real dialogue.
- Utterance-Single: The manipulation of data is only applied once to a random utterance in a real dialogue.
- Mixed-Single: The manipulation of data is only applied once to a random utterance or a random segment no longer than 20 tokens in a real dialogue.
- Mixed-Multi: The manipulation of data is applied randomly for a uniformly random value from 1∼4 times.
- AMR-Multi: The location of data manipulation is guided by similar approach as in DEAM Ghazarian et al. (2022).

Similarly, we study following aggregation strategies for discriminative CDM:

- Pooling along the timestep axis (Avg-Pooled/Max-Pooled/Min-Pooled)
- Classifier-Pooled: We train a small linear classifier to convert the sequence of contrastive momentums as a trainable pooler using annotated training data (from the original dataset or as synthesized by DEAM(Ghazarian et al., 2022)).

### 4.1.2 RESULTS AND ANALYSIS

We present quantitative results in Table 2. Given that CDM is designed to model discrete comparison relations, our approach aligns with methodology established by previous research (Mesgar et al., 2019; Vakulenko et al., 2018; Zhang et al., 2021; Ghazarian et al., 2022) and report the Spearman correlation to better evaluate CDM against these baselines.

Table 2: Spearman correlation of different approaches for dialogue evaluation. All reported correlation coefficients from our approaches have $p$-value (with Bonferroni correction) < 1e-2. We highlight the best-performing results with **bolded** numbers and second-best with underlined numbers. The models are selected using the validation set of Topical-Personal chat.

| Model | FED | | DSTC9 | |
| --- | --- | --- | --- | --- |
| | Coherence | Overall | Coherence | Overall |
| Mesgar et al. (2019) | 0.10 | -0.01 | 0.02 | 0.05 |
| Vakulenko et al. (2018) | 0.13 | 0.10 | 0.00 | 0.00 |
| DynaEval (Zhang et al., 2021) | -0.36 | -0.4 | -0.03 | -0.01 |
| DEAM (Ghazarian et al., 2022) | 0.47 | 0.55 | 0.19 | 0.20 |
| Generative CDM (Ours) | 0.51 | 0.52 | 0.19 | 0.23 |
| Discriminative CDM (Ours) | **0.59** | **0.61** | **0.27** | **0.25** |

We find that DSTC9 favors segment-level manipulation while FED favors utterance-level manipulation. Additionally, our findings indicate that larger performance gap between the amateur/expert models in general induces better performance. Finally, Discriminative CDM methods present less bias across datasets and offer more efficiency during training, as they eliminate the necessity for training an additional classifier model.

## 4.2 FACTUALITY EVALUATION FOR ABSTRACTIVE SUMMARIZATION MODELS

We now concern a slightly different setup, where we use CDM for evaluating the factuality of abstractive summarization models. Previous methods usually consider the problem as an NLI problem with only the "*entailment*" case of annotations. We evaluate our model in comparison to existing works, such as Falsesum(Utama et al., 2022), QAGS(Wang et al., 2020), Coco(Xie et al., 2021) and FactCC(Kryściński et al., 2019).

Table 3: Ablation study on how different variants and hyperparameters of the proposed CDM impact the final performance of the methods.

| Model | FED | | DSTC9 | |
|---|---|---|---|---|
| | Coherence | Overall | Coherence | Overall |
| Generative CDM | | | | |
| - Segment-Single (small-large) | 0.12 | 0.07 | 0.11 | 0.10 |
| - Utterance-Single (small-large) | 0.29 | 0.36 | 0.05 | 0.08 |
| - Mixed-Single (small-large) | 0.32 | 0.35 | 0.14 | 0.12 |
| - Mixed-Multi (small-large) | 0.42 | 0.40 | 0.17 | 0.18 |
| - AMR-Multi (small-large) | 0.49 | 0.53 | 0.20 | 0.22 |
| - AMR-Multi (small-base) | 0.48 | 0.51 | 0.19 | 0.20 |
| - AMR-Multi (small-large) | 0.49 | 0.53 | 0.20 | 0.22 |
| - AMR-Multi (small-xl) | 0.51 | 0.52 | 0.19 | 0.23 |
| Discriminative CDM | | | | |
| - Avg-Pooled (small-large) | 0.31 | 0.32 | 0.12 | 0.13 |
| - Min-Pooled (small-large) | 0.27 | 0.28 | 0.07 | 0.04 |
| - Max-Pooled (small-large) | 0.46 | 0.43 | 0.16 | 0.15 |
| - Classifier-Pooled (small-large) | 0.53 | 0.56 | 0.24 | 0.22 |
| - Classifier-Pooled (small-base) | 0.42 | 0.44 | 0.13 | 0.10 |
| - Classifier-Pooled (small-large) | 0.53 | 0.56 | 0.24 | 0.22 |
| - Classifier-Pooled (base-large) | 0.39 | 0.40 | 0.09 | 0.11 |
| - Classifier-Pooled (small-3b) | **0.59** | **0.61** | **0.27** | **0.25** |

### 4.2.1 DATASET AND EXPERIMENT SETUP

We adopt most experimental settings from existing works on factuality evaluation of abstractive summarization. For hyperparameter search and early stopping purposes, we randomly select 5% of the training data as the development split. The following table lists some key statistics of the datasets we use in this part of experiments.

Table 4: Data usage in our experiments

| Dataset | size |
|---|---|
| CNN/DailyMail (train) (Nallapati et al., 2016) | 1,003,355 |
| QAGS-CNN/DailyMail (test, w/ human annotation) (Wang et al., 2020) | 235 |
| QAGS-XSum (test, w/ human annotation) (Wang et al., 2020) | 239 |
| SummEval (test, w/ human annotation) (Fabbri et al., 2021) | 1200/3600 |

We train our likelihood functions under both supervised and unsupervised setups:

- **Unsupervised** For annotated data, we only train the likelihood function to *maximize* the likelihood of positive samples. In this sense, the model is more similar to the one we used in dialogue evaluation.

- **Supervised** We train the likelihood function as factual-counterfactual label-conditioned probabilities using synthesized pseudo labels following the methods as described in FalseSum(Utama et al., 2022). When performing the discriminative CDM, we use factual-conditoned larger model as the *expert* model and counterfactual-conditioned smaller model as the *amateur*.

- **Cross-Supervised** This setup is almost identical to **Supervised**, except that we're using an extra Generative CDM as the negative sampler other than FalseSum. This shows that the performance improvements of **Supervised** is not majorly coming from implicit ensemble with FalseSum.

### 4.2.2 RESULTS AND ANALYSIS

We show results in Table 5. Previous works report their performance inconsistently in either Spearman/Pearson correlation or an accuracy score with 0/1 quantization of the annotations. We adopt 0/1-quantization and report the accuracy of each baseline/model.

Table 5: Evaluation results for abstrative text summarization. For all models and datasets, we show the quantized accuracy scores. Results with * means ones from our re-implementation or re-evaluation so as to unify the metric.

|  | SummEval | QAGS Overall | QAGS CNN/DM | QAGS XSum |
|---|---|---|---|---|
| Falsesum (Utama et al., 2022) | 65.18 | 75.05 | **94.89*** | 55.65* |
| QAGS (Wang et al., 2020) | 59.82* | 72.15 | 88.08* | 56.48* |
| Coco (Xie et al., 2021) | 66.71* | 77.00* | 93.62* | 60.66* |
| FactCC (Kryściński et al., 2019) | 60.04 | 73.42 | 85.96* | 61.09* |
| Generative CDM (Ours, small-large, Segment-Multi) | 65.35 | 76.16 | 91.48 | 61.08 |
| Discriminative CDM (Ours, small-xl, Cross-Supervised) | **68.17** | **78.27** | 92.34 | **64.43** |

For an ablation study of different training strategies of Discriminative CDM and the corresponding discussion, please refer to the appendix.

## 5 CONCLUSION AND FUTURE WORK

This paper presents the Contrastive Distribution Methods (CDM) as a general framework for evaluating open-domain text generation models. CDM is constructed around analyzing the correlation between model scales and the respective distribution prediction, and how it can be exploited to alter the performance of a certain model on-the-fly in inference. We demonstrate how CDM can be used for evaluation purposes in two general paradigms: Generative CDM, which manipulates existing positive samples to generate in-domain negative samples and subsequently trains a classifier, and Discriminative CDM, which employs the contrastive momentum as a direct metric for evaluation. Our experiments results in multi-turn dialogue evaluation and factuality evaluation for abstractive summarization illustrate that CDM correlates better with human intuition than traditional metrics. In summary, the CDM method emerges as a promising and scalable approach for evaluating open-domain text generation systems, among others.

For future work, it is interesting to consider the contrastive distribution concerning more than two distributions as a reflection of an extended series of models across different scales. Furthermore, it presents an interesting avenue to explore whether the Generative CDM approach can be extended for a more effective ensemble of heterogeneous models that differ in scale, architecture or even training data but operate under a reasonable assumption of a partially ordered performance level.

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

# A  APPENDIX

## A.1  ALGORITHM DESCRIPTION

The full process of generative CDM can be summarized as follows:

---
**Algorithm 1** Generative CDM
---
1:  Train the amateur model $p_a$ to solve the segment insertion problem
2:  Train the expert model $p_e$ to solve the segment insertion problem
3:  Construct the contrastive momentum $m_{a \to e} = \log p_e - \log p_a$
4:  Construct the degraded distribution $\log p_a^- \propto \log p_a - \gamma m_{a \to e}$
5:  negativeSamples = { }
6:  **for** positiveSample $\mathbf{s}^+$ **in** positiveSamples **do**
7:      Remove a segment $\mathbf{e}^+ \subset \mathbf{s}^+$ from $\mathbf{s}^+$ to construct the context $\mathbf{c} = \mathbf{s}^+ - \mathbf{e}^+$
8:      Regenerate a segment $\mathbf{e}^-$ in the same position using $p_a^-(\mathbf{e}|\mathbf{c})$
9:      Obtain the reconstructed negative sample $\mathbf{s}^- = \mathbf{c} \cup \mathbf{e}^-$
10:     Add $\mathbf{s}^-$ to negativeSamples
11: **end for**
12: Train the metric model $D$ as a discriminator with {negativeSamples, positiveSamples}
13: **return** metric $D$

---

## A.2  ABLATION STUDY OF THE COMPONENT CHOICES IN DISCRIMINATIVE CDM

Table 6: Ablation study on different training strategies of Discriminative CDM on abstractive summarization generation tasks.

| | SummEval | QAGS Overall | CNN/DM | XSum |
|---|---|---|---|---|
| Discriminative CDM (Ours) | | | | |
| - Max-Pooled (small-large, Unsupervised) | 64.17 | 74.05 | 85.11 | 63.18 |
| - Classifier-Pooled (small-large, Unsupervised) | 64.74 | 74.47 | 90.21 | 59.00 |
| - Classifier-Pooled (small-large, Supervised) | 67.17 | 76.79 | 93.19 | 60.67 |
| - Classifier-Pooled (small-large, Cross-Supervised) | 66.59 | 76.79 | 92.34 | 61.50 |
| - Classifier-Pooled (small-xl, Supervised) | **68.17** | **78.27** | 93.62 | 63.18 |
| - Classifier-Pooled (small-xl, Cross-Supervised) | **68.17** | **78.27** | 92.34 | **64.43** |

### A.2.1  DISCUSSION

- Learned pooling does not show as significant improvements as in dialogue for factuality evaluation. This is probably due to that the majority of the output semantics is already contained within the condition, and the statistical properties of the step-wise contrastive momentum score are more consistently distributed.

- With the explicitly annotated data with pseudo labels, using the supervised setting is in general performing better. There could be multiple potential explanations:

  - First, conditioning the likelihood function on the pseudo labels implicitly creates an ensemble of CDM and the existing method Falsesum (Utama et al., 2022). The potential improvement can majorly be a result of such implicit ensemble. However, results from the *Cross-Supervised* variant of the model disprove this possibility.

  - Thus, we argue this might due to that using supervised data and label-conditioning the probability function helps to enlarge the discrepancy between the *amateur* and *expert* models, which helps to generate a better, generalize decision boundary for CDM.

- Compared to **Supervised** setup, **Cross-Supervised** setup slightly favors the *XSum* subset (slightly out of domain) of the QAGS dataset over *CNN/DM* (mostly in-domain), but generally performing at a similar level. We argue that this shows that the decision boundary learned by CDM could be more robust towards domain-shift.

## A.3 LIMITATION

We hereby list a few potential limitations of the proposed method:

- While the method is proposed as a very general framework, training a metric with CDM is still a highly task-dependent practice. While this paper has found several practical principles (enlarging the discrepancies between models, etc), it could still need some efforts to apply CDM to a new task domain.

- It is theoretically infeasible for CDM to produce metrics for evaluating the inverse-scaling tasks (bigger the base model, worse the performance), as it goes against the very basic assumption the CDM approaches rely on.

- Currently presented results of CDM are based on the linear approximation of the oracle $E(p)$ using a secant hyperplane. This approach can be highly limited compared to a more accurate approximation. We leave this for future work.

## A.4 EXTENSIVE STUDY OF DIFFERENT MODELS

Due to the capacity limit in the main text, we haven't been able to include a larger range of study on how different models perform as the amateur/expert. We hereby include these results in Table 7.

The prompts (for overall/coherence evaluation) we used for G-Eval are adapted from the original G-Eval repository:

> You will be given a conversation between two agents.
>
> Your task is to rate the dialogue on one metric.
>
> Please make sure you read and understand these instructions carefully. Please keep this document open while reviewing, and refer to it as needed.
>
> Evaluation Criteria:
>
> Overall (1-10) - the overall quality of the whole dialogue.
>
> Evaluation Steps:
>
> 1. Read the dialogue carefully to get a general understanding of the overall quality of it.
>
> 2. Assign an overall score on a scale of 1 to 10, where 1 is the lowest and 10 is the highest based on the Evaluation Criteria.
>
> Dialogue:
>
> {**The Dialogue**}
>
> Evaluation Form (scores ONLY):
>
> - Overall:

> ......
>
> Evaluation Criteria:
>
> Coherence (1-10) - the collective quality of all utterances.
>
> Evaluation Steps:
>
> 1. Read the dialogue carefully and identify the main topic and key points.
>
> 2. Read the utterances and compare it to the previous ones. Check if the utterance stays on topic and maintains key points of the dialogue, or performs a smooth transition between different sub-topics.
>
> 3. Assign a score for coherence on a scale of 1 to 10, where 1 is the lowest and 10 is the highest based on the Evaluation Criteria.
>
> ......

Table 7: Spearman correlation of approaches studied in a larger range for dialogue evaluation. All reported correlation coefficients from our approaches have $p$-value (with Bonferroni correction) < 1e-2.

| Model | FED | | DSTC9 | |
|---|---|---|---|---|
| | Coherence | Overall | Coherence | Overall |
| **Generative CDM** | | | | |
| - AMR-Multi (small-base) | 0.48 | 0.51 | 0.19 | 0.20 |
| - AMR-Multi (small-large) | 0.49 | 0.53 | 0.20 | 0.22 |
| - AMR-Multi (small-xl) | 0.51 | 0.52 | 0.19 | 0.23 |
| - AMR-Multi (small-11b) | 0.53 | 0.55 | 0.22 | 0.24 |
| **Generative CDM (Resampling using only amateur)** | | | | |
| - AMR-Multi (small) | 0.31 | 0.28 | 0.09 | 0.08 |
| - AMR-Multi (base) | 0.19 | 0.16 | 0.05 | 0.04 |
| - AMR-Multi (large) | 0.09 | 0.05 | -0.01 | 0.02 |
| **Generative CDM with Pythia-standard** | | | | |
| - Utterance-Multi (70M-160M) | 0.26 | 0.27 | 0.02 | 0.04 |
| - Utterance-Multi (70M-410M) | 0.31 | 0.29 | 0.04 | 0.04 |
| - Utterance-Multi (70M-1.0B) | 0.09 | 0.05 | -0.01 | 0.02 |
| **Generative CDM with Pythia-standard, w/ Infilling Objective** | | | | |
| - AMR-Multi (70M-160M) | 0.28 | 0.31 | 0.04 | 0.06 |
| - AMR-Multi (70M-410M) | 0.32 | 0.34 | 0.09 | 0.10 |
| - AMR-Multi (70M-1.0B) | 0.35 | 0.37 | 0.10 | 0.11 |
| - AMR-Multi (70M-1.4B) | 0.35 | 0.38 | 0.10 | 0.12 |
| - AMR-Multi (70M-6.9B) | 0.44 | 0.41 | 0.18 | 0.22 |
| **Finetuned LLaMa 2, No Infilling** | | | | |
| - Generative CDM (Utterance-Multi, 7B-13B) | 0.18 | 0.21 | 0.06 | 0.08 |
| - Discriminative CDM (Utterance-Multi, 7B-13B) | 0.20 | 0.22 | 0.13 | 0.15 |
| **Discriminative CDM** | | | | |
| - Avg-Pooled (small-large) | 0.31 | 0.32 | 0.12 | 0.13 |
| - Min-Pooled (small-large) | 0.27 | 0.28 | 0.07 | 0.04 |
| - Max-Pooled (small-large) | 0.46 | 0.43 | 0.16 | 0.15 |
| - Classifier-Pooled (small-large) | 0.53 | 0.56 | 0.24 | 0.22 |
| - Classifier-Pooled (small-base) | 0.42 | 0.44 | 0.13 | 0.10 |
| - Classifier-Pooled (small-large) | 0.53 | 0.56 | 0.24 | 0.22 |
| - Classifier-Pooled (base-large) | 0.39 | 0.40 | 0.09 | 0.11 |
| - Classifier-Pooled (small-3b) | 0.59 | 0.61 | 0.27 | 0.25 |
| G-Eval (w/ gpt4-0613) | **0.72** | **0.73** | **0.28** | **0.27** |
| UniEval (w/ Avg.Pool) | 0.33 | 0.35 | 0.14 | 0.13 |

**Key Updates**

- We include some more recent metrics like UniEval(Zhong et al., 2022b) and G-Eval (Liu et al., 2023a). Note that for dialogue evaluation, the presented UniEval can only serve as an utterance-level evaluator. To extend its ability, we have to perform an average pooling over each responded utterance to collect the final score.

- We include results more recent and larger language models like Pythia(Biderman et al., 2023) and LLaMa 2(Touvron et al., 2023b). We apologize that we currently are only able to afford to present results with at most as large as 13B-sized variants of such models. We plan to include the LoRA-finetuned versions of larger variants in the camera-ready version. Please note that, unlike T5 (Raffel et al., 2019), these two models are never pretrained to perform infilling, so many of the setups that can naturally work for T5 models may not be directly applicable for these two models.

- We include Generative CDM results using only the amateur model to serve as the ablation study.

- We extend our discussion on the ablation study by further extending the range of T5 models to as large as 11b. Note that even if T5 is an old model, the larger versions of it is still very capable and comparable with many more recent open-sourced language models.

**Discussion**

- G-Eval is no doubt the state-of-the-art among all metrics studied. Yet we argue this is less of a fair comparison, as gpt4(OpenAI, 2023) is explicitly trained to align with human opinions. Our contribution focuses on studying the scaling dynamics of language models and how evaluation using such dynamics can align reasonably well with human intuition without explicitly training the model to do the alignment. Since it is still encourage to report non-SOTA yet independent and contributing ideas, we believe our contribution in this paper is still valuable.

- UniEval w/ Average Pooling performs very similar to DiscCDM with the same pooling strategy. We argue that this can be a direct result of some deeper connection and latent mechanism between the two methods. We leave further investigation for future work.

- The conducted ablation study yields a few principled practice in CDM:
  - First, we in general want to enlarge the discrepancy between the amateur model and expert model to achieve better performance for both Generative and Discriminative CDM.
  - Second, Discriminative CDM w/ a trainable, linear time series model-based pooler is in general less sensitive to the discrepancy between amateur and expert models. It's also showing better performance in most tasks.
  - Second, in GenCDM, models with infilling capabilities (conditioned on bidirectional context) usually serve as a better negative sampler compared to autoregressive models that can only manipulate the post-text given the pre-text. Specifically, models are pretrained to perform infilling performs even better than those autoregressive models which are never pre-trained but only finetuned to perform infilling.

- Directly sampling from the amateur model is not working. Sampling from a stronger amateur model alone yields a worse metric. This is understandable, as ideal negative samples should be those that are with high probability in machine perception but lower quality by human opinion. When the model sizes increase, the manipulated samples can be undistinguishable from the golden samples, making the training of the classifier-based metric either impossible or simply overfitting the noise.

- Compared to model with infilling capability, autoregressive models that only manipulate/predict post-text using pretext performs worse. This is rather understandable in Generative CDM, but the consistent finding in Discriminative CDM (while less severe) is actually interesting. We assume that using birectional perception in such evaluation process can implicitly amplify the factor towards machine-generated flaws within the context, since it both **negatively contribute to the scores of itself** and **any closely correlated context**. This especially helps when we try to pool the scores from multiple tokens and sentences into a dialogue-level score, making the resulting metric sharper against a local artifact. We leave the investigation for evidence to prove/disprove such assumption for future work.

