# OpenReview forum: "Open-Domain Text Evaluation via Contrastive Distribution Methods"
_ICLR.cc/2024/Conference — Submitted to ICLR 2024_

### Official Review · Reviewer_qwhh · 2023-10-29

**Soundness:** 3 good
**Presentation:** 3 good
**Contribution:** 2 fair
**Rating:** 5
**Confidence:** 3

**Summary:**

The paper presents a novel method, called "contrastive distribution methods (CDM)," for assessing open-domain text generation. CDM offers two distinct evaluation metrics. First, the generative CDM focuses on the creation of negative samples for training a classifier. Second, the discriminative CDM emphasizes calculating the probability of sequence generation using a contrastive approach. Experimental results demonstrate the effectiveness of both the generative and discriminative CDM when comparing to several baseline methods.

**Strengths:**

**Clarity**: The paper is well written and easy to follow. It begins by outlining the assumption made (i.e., larger models perform better than smaller models at generation tasks), followed by a detailed explanation of the two CDM methods derived from the assumption. The experimental sections are clearly presented, offering comprehensive details and ablation studies.

**Originality**: The paper is original in the sense that it applies the contrastive decoding idea into developing both generative CDM and discriminative CDM.

**Substance**: Comprehensive ablation studies are carried out in the paper. These studies effectively determine the optimal strategies for constructing negative samples in the generative CDM and for aggregating in the discriminative CDM.

**Weaknesses:**

**Soundness**:
The paper appears to overlook crucial baselines. Specifically:
- For the generative CDM, there's no direct comparison with the expert/amateur model employed solely for constructing negative samples.
- For the discriminative CDM, a direct comparison is missing where the expert model is used to compute the generation score.

These baselines are essential for establishing the paper's originality (applying the contrastive decoding idea into text generation evaluation) and shouldn't be omitted.

**Limitations**:
The paper acknowledges some limitations in Appendix A.3. Additionally, I'm concerned regarding the method's applicability in two trends of open-domain generation evaluation:
   - **Fine-grained Evaluation**:  This is the case where users arbitrarily set the evaluation perspective. It's unclear if the negative sample creation strategy can consistently control construction towards a specific aspect.
   - **Holistic Evaluation**: This involves comparing generated outputs to a given input and assigning ranks. Given that the paper mentions that state-of-the-art Language Models like LLAMA-1/2 7b are too strong to act as amateur models, it suggests the approach is best suited for weaker models. This raises the question: can it effectively evaluate outputs from more advanced models? This potential constraint could limit its applicability.

**Questions:**

See the limitations in weaknesses.

---

> ### Author Response · Authors · 2023-11-22
> **Response to qwhh**
>
> We appreciate the reviewer's suggestions and comments. We've updated our draft to include some extra results and discussions in the Appendix. We hereby respond to the questions and comments one-by-one.
>
> 1. (Soundness: Overlooked baselines) Thank you. We've conducted extensive study and reported the results in the updated draft's appendix. Please refer to Appendix A.4, Table 7.
>
> 2. (Limitations: Fine-grained Evaluation) We admit that this is indeed the case for Generative CDM. For Discriminative CDM, fine-grained evaluation can be achieved by re-aligning the score via training the learnable pooler on a small set of samples that illustratively defines the fine-grained aspect.
>
> 3. (Limitations: Holistic Evaluation) This is an insightful comment. We thank the reviewer for pointing this out to us. One of our future works is exactly on this direction, to study the multi-model (>2) case of CDM, to train a CDM system that consists of a series of rather small models (15M to 1.5B) and whether it can extrapolate the decision surface to correctly ranking stronger models (3B, 6.9B, 7B etc) accordingly to their performance. For this paper, though, we tend to keep only the core demostration and verification of effectiveness in a comparitively simplified setup. We respectfully beg for understanding from the reviewer.

---

### Official Review · Reviewer_6ryG · 2023-10-31

**Soundness:** 2 fair
**Presentation:** 3 good
**Contribution:** 3 good
**Rating:** 5
**Confidence:** 3

**Summary:**

The paper introduces a new framework, Contrastive Distribution Methods (CDM), for evaluating open-domain text generation. CDM is based on the assumption that larger model tends to have better open-domain generation performance than smaller ones, and is done by mapping the contrasts of two probability distributions to a quality measure. The authors explored two paradigms of CDM: Generative CDM, which generates synthetic examples for training a classifier, and Discriminative CDM, which uses the contrast between two distributions for direct evaluation. Experimental results show that CDM improves correlation with human judgments in multi-turn dialogue evaluation and factuality evaluation for abstractive summarization.

**Strengths:**

- The research problem (open-ended text evaluation) is important for current NLP community.
- The experiment results look promising.
- The proposed method can fit in both the generative and discriminative paradigms explored in the past years.

**Weaknesses:**

- The experiments were only conducted on T5 models, which raise the question of how well the decoder-only models perform with CDM.
- The proposed method relies heavily on the divergence between two distributions for quality prediction and it seems (1) picking the right amateur model may require extra effort, and (2) whether the method generalizes to larger model scales (e.g., >7b) is unknown. As noted by the authors, “Note that the smallest configurations… such as LLaMa-1/2(Touvron et al., 2023a;b), can still be too strong to serve as the amateur model in CDM”

**Questions:**

- Have you tried switching both amateur and expert models to larger scales? Will contrasting 7B and 70B output distribution yield good correlation with humans?
- Have you tried any decoder-only language models?

---

> ### Author Response · Authors · 2023-11-22
> **Response to 6ryG**
>
> We thank the reviewer for the mild criticisms and valuable suggestions. We've updated our draft to include some extra results and discussions in the Appendix. We hereby respond to the questions and comments one-by-one.
>
> Re Weaknesses:
>
> 1. (More Experimented Models) Thank you. We've conducted extensive study and the results can be found in Appendix A.4. Decoder-only model generally suffers from the problem that it can only make use the pre-text to manipulate/predict the post-text, limiting its application in both Generative CDM and Discriminative CDM. But we show that this can be slightly remedied by finetuning autoregressive decoder-only language models to perform infilling, following the practice described in https://arxiv.org/abs/2207.14255 . Yet this remedy is not sufficient to essentially close the gap to those models that are pretrained to perform infilling, like T5.
>
> 2. (Regarding selection of Amateur Models) We find that in general using a smaller amateur is always helpful, and in the extensive study shown in the appendix, we find that when the expert model is as large as 3b or 11b, the performace of CDM on the dialogue evaluation problem is already coming to a saturation. We also conduct experiments using LLaMa 2 7b against 13b, and find that a 7b model is indeed too strong as an amateur model.
>
> Re Questions:
>
> 1. (Larger Scales of Models) We've included some results contrasting T5-small against T5-11b, and also results from LLaMa 2 7b against 13b. Unfortunately we can't afford to adapt LLaMa2 70b for our purpose. We beg the reviewer's pardon on this.
>
> 2. (Decoder-only models) Yes, we've included results from Pythia and LLaMa 2 in the extensive study in Appendix A.4. In summary, decoder-only language models are less ideal for CDM in multiple aspect. Detailed results, discussion and analysis can be found in the updated draft.

---

### Official Review · Reviewer_kEEE · 2023-11-03

**Soundness:** 3 good
**Presentation:** 3 good
**Contribution:** 3 good
**Rating:** 6
**Confidence:** 3

**Summary:**

The authors introduce a reference-free evaluation metric for open-ended text generation. It exploits the properties between small amateur and large expert models; in the paper T5 of varying sizes is used. Results are compelling on multiple datasets, both for dialogue and summarization; summarization measures factual consistency between the input and output. The paper is most lacking in analysis to further convince the reader the metric is capturing the desired properties.

**Strengths:**

1a. The method is clever, effective, and mostly straightforward. It exploits contrastive properties of small amateur models and large expert models. Perhaps others can build upon the contrastive nature of this setup for improved or other types of evaluation.

1b. It is actually two methods, and the classifier-based discriminative method seems particularly effective.

2. Promising results on multiple datasets, correlating outputs with human preferences. Including results on factual consistency.

3. Includes ablation study on model sizes and other settings.

**Weaknesses:**

1. Performance seems especially bound by the expert model. This is not always the case for reference-free evaluation. Perhaps it is worth analyzing the cases where the metric fails.

2. Although main results on multiple tasks are promising, the paper lacks analysis (qualitative or quantitative) to convince the reader CDM is capturing desirable properties of the outputs.

3. Negatives in generative CDM are created at the segment level. My impression is this would lead to grammatically similar but semantically different negatives. Stronger negatives would likely differ in style and structural complexity. Even basic ordering differences, such as swapping "A is a B" with "B is a A" are not likely to be handled.

4. The metric is evaluated solely against human preferences. It would be helpful to see if the metric can properly order a set of baseline models by their relative strength. Although it's possible I may have misunderstood if this is already being done or not.

**Questions:**

Q1: Is it really a safe assumption that small is worse than large? Aren't larger models also harder to train? I understand in practice your choice of T5 probably does obey this pattern, but naively scaling a model up or down and keeping most of the hyperparams fixed may not follow this pattern.

Q2: Can we do paraphrasing instead of segment replacement?

Q3: Did you consider using multiple seeds of amateur/expert? Do you expect it to influence results?

Q4: Have you considered using multiple contrastive evaluations combined? For instance, could train particularly biased amateur. For inspiration, see "He He et al. Unlearn Dataset Bias in Natural Language Inference by Fitting the Residual"

---

> ### Author Response · Authors · 2023-11-22
> **Response to kEEE**
>
> We thank the reviewer for the mild appreciation of our work. We've updated our draft to include some extra results and discussions in the Appendix. We hereby respond to the questions and comments one-by-one.
>
> Re Weaknesses:
> 1. (Performance bound by the expert model) Yes, this is our finding too. In the extensive study we further extend the size of the largest studied T5 model to as large as 11b and we observe that the performance seems to be saturating.
>
> 2. (Lacking Analysis and/or Illustrative Examples) Thank you for this suggestion. Due to the time constraint and paper length capacity, we're not able to incorporate more illustrations and detailed qualitative analysis into the draft. We're continuously working on this and more results will be present in the camera ready version.
>
> 3. (Negative Sampling Generation) We respectfully disagree with the reviewer that we're only doing manipulation in the segment level. We've conducted ablation study to compare the performance of doing negative sampling in segment, utterance and guided positions using AMR, and do the manipulation for 1 steps or multiple steps. Please refer to Section 4, Table 3 and Appendix, Table 7.
>
> 4. (Metric Alignment and Partial Order of Models) We thank the reviewer for this suggestion. This is definitely interesting and we're doing it as one of our follow-up works of this paper. As described in the future work section, this follow-up work will focus on the multi-model version of CDM, and we're testing its capabilities of ranking models accordingly to their respective capabilites.
>
> Re Questions:
>
> 1. (Whether the partial order assumption generally holds) We definitely agree that this should be the case. We stay agnostic of whether this assumption generally holds for any aspect of text quality, but we tend to believe this assumption generally holds for a considerably large enough range of aspects. Our method is trying to build towards cases where the assumption safely holds.
>
> 2. (Paraphrasing against Segment Replacement) This is insightful and actually links to some previous results in edit-based text style transfer. To summarize, these results indeed show that when the target is to accurately reflect the opposite information of the current context, doing sentiment-reversed paraphrasing is indeed more effective. However, our case here is a little bit different - we don't actually care too much about preserving the semantics of the manipulated region. Instead, we only care what could be properly substituting the deleted part that best "confuses" a potentially weaker discriminator while still making the full text a negative sample. We do think it's possible, but the formulation and training will be more complex and we doubt the performance gain in this negative sample synthesis problem.
>
> 3. (Random Seeds Issue) Yes, we run the main results using different random seeds for 10 times and make sure the conclusion holds.
>
> 4. (Intentionally Manipulated Amateur Model) This is possible, but it could slightly deviate from the paper's main focus on showing that "quantification of LM scaling law can be used for sequence-level evaluation purposes".

---

### Official Review · Reviewer_V2uz · 2023-11-09

**Soundness:** 3 good
**Presentation:** 2 fair
**Contribution:** 2 fair
**Rating:** 5
**Confidence:** 4

**Summary:**

The paper proposes an interesting Contrastive Distribution Methods (CDM) for text generation evaluations. The idea presented in the paper is highly influenced from the contrastive decoding (Li et al. 2022) where the model tries to decode a sequence maximizing the contrastive momentum: m(s) = log p_expert(s) - log p_amateur(s). It builds on the partial order assumption that  the model with a larger number of parameters (expert) performs better than the smaller one (amateur), in the same model family. The paper explores two ways to use contrastive momentum for text evaluation: 1) generative CDM: the constravie momentum is used to generate negative examples by perturbing positive examples and then a discriminator is trained to separate positive examples from negative examples, and 2) discriminative CDM: estimated as the sum of the contrastive momentum at each time step.

The experiments are done with multi-turn dialog (focusing on overall and coherence quality) and summarization (focusing on factuality) evaluations.  Discriminative CDM seems to perform better than the Generative CDM approaches on both tasks.

The paper is very interesting to read until the experiment section. The experiments and results section could have been better to strengthen their conclusions regarding their proposed methods for text evals.

**Strengths:**

The use of contrastive momentum for text evaluation is very interesting and could be easily generalized to different tasks and languages.

The experiments are done on multi-turn dialogue and factuality in abstractive summarization, showing positive results for CDM. The authors have also investigated different ways of perturbing positive examples with CDM.

**Weaknesses:**

I felt that the experiment and results in the paper could have been a bit more thorough to make a strong conclusion about CDM as a general metric for text evaluation. Lots of questions were left unanswered. It will make it hard for people to adapt CDM in their usecases. I elaborate those questions below in the Question section.

**Questions:**

I am not certain if the partial order assumption holds for different aspects of the text quality. For example, the authors evaluate CDM for the overall quality and coherence for dialog and factuality for summarization. It would be interesting to see if the partial order assumption holds for factuality, for examples, in the first place. The authors should report performance of expert and amateur models on different aspects.

The authors have tried various sizes of T5 models. I think it might be of interest to include other model families in their comparisons.

For summarization, why are the authors focusing on factuality only? What about other dimensions: for example, Coherence,  Factuality,  Fluency,  Informativeness? It would have been nice to see how CDM does on various aspects. Also the comparison is limited to very few other metrics. Please see https://openreview.net/forum?id=OIe3kpwl40D for a better experimental setup.

The captions in Table 1 and Table 4 are not very clear. Is the data in the first block used to train T5 or do we use off-the-shelf T5 checkpoint? What is the “/” in 1200/3600 and 17567/2078?

“Previous works report their performance inconsistently in either Spearman/Pearson correlation or an accuracy score with 0/1 quantization of the annotations. We adopt 0/1-quantization and report the accuracy of each baseline/model.” → This is not at all clear. It would be good to clarify this.


Minor:

Section 4.2.2: Table 6 -> Table 5

sum_log m -> sum_m in Section 3.4.2.

---

> ### Author Response · Authors · 2023-11-22
> **Response to V2uz**
>
> We thank the reviewer for the detailed questions and valuable comments. We've updated our draft to include some extra results and discussions in the Appendix.
>
> 1. (Whether the partial order assumption generally holds) We definitely agree that this should be the case. We stay agnostic of whether this assumption generally holds for any aspect of text quality, but we tend to believe this assumption generally holds for a considerably large enough range of aspects. Our method is trying to build towards cases where the assumption safely holds.
>
> 2. (Other model families) Thank you, we've conducted further experiment on a larger range of models, including larger sizes of T5 and decoder-only autoregressive models (Pythia, LLaMa 2). Due to the constrained time, we've only been able to present the extensive results on the dialogue evaluation data, but we plan to continuously work out all results on summarization too before the camera ready.
>
> 3. (Experimental Setup regarding summarization) We focus on factuality evaluation because we find this could be a challenging aspect for abstractive summarization models, following a few previous works like QAGS, FactCC. We agree that it is definitely valuable and interesting to extend the metric learning to more aspects, but we beg the reviewer's pardon as we struggle to fit into the paper capacity limit.
>
> 4. (Data Table Issues) Yes, as is indicated by the information in the parentheses, we are training amateur and expert models on the CNN/DailyMail data and testing the composed CDMs on QAGS and SummEval datasets. "/" indicates different sub-sets of the data (TopicalChat: 17567 conversations; PersonalChat: 2078 conversations; 1200 summaries in SummEval with 3600 annotated factuality scores by CoCo https://aclanthology.org/2021.findings-emnlp.10.pdf ).
>
> 5. (Result Reformatting Issues) Some previous works report their performance by treating factuality evaluation as a balance-labelled binary classification problem and report the accuracy (FactCC, FalseSum, QAGS), whereas some report the Spearman and Pearson Correlation (CoCo). To provide a clearer comparison, we tried to reproduce each of the compared baselines and unify the results into the accuracy measure. We binarize CoCo's results into discrete True/False prediction using a threshold determined by top50% of the CoCo score, as is described in their paper.
>
> 6. (Minor Writing Issues) Thank you. We've fixed them accordingly.

---

### Official Review · Reviewer_CKwf · 2023-11-10

**Soundness:** 2 fair
**Presentation:** 2 fair
**Contribution:** 2 fair
**Rating:** 3
**Confidence:** 4

**Summary:**

This paper introduces a new text evaluation method called Contrastive Distribution Methods (CDM). Among these, Generative CDM harnesses the contrast of two language models’ distributions to generate synthetic examples for training discriminator-based metrics, while  Discriminative CDM directly uses distribution disparities between two language models for evaluation. Experiments show the effectiveness of CDM on multi-turn dialogue and factuality in abstractive summarization.

**Strengths:**

1. The proposed method can outperform several baselines on both dialogue and summarization evaluation tasks.

**Weaknesses:**

1. The method design of negative sample generation seems not to make sense for me. It is commonly known that high-quality negative samples may promote the performance of discriminators. Even if low-quality samples are needed, large language models (LLMs) such as GPT-4 also have the ability to generate texts with controlled qualities via prompt design. Thus, I’m curious about the necessity to train a model on task-specific data to construct negative samples, which may degrade the generalization ability of the proposed method.

2. The baselines used in the experiment are somewhat outdated with the rapid development of LLM-based evaluation metrics. Since the authors have already cited two representative works including UNIEVAL [1] and G-EVAL [2] in the introduction and claim that they have limitations, these two methods should be surely added to the baselines for direct comparison.

3. The experimental analysis in this paper is too rough and needs to be largely improved. I even don’t find the analysis on important ablation studies in the main content.

[1] Towards a Unified Multi-Dimensional Evaluator for Text Generation. EMNLP 2022.

[2] G-Eval: NLG Evaluation using GPT-4 with Better Human Alignment. arXiv 2023.

**Questions:**

I have included my questions in the weaknesses part.

---

> ### Author Response · Authors · 2023-11-22
> **Response to CKwf**
>
> We thank the reviewer for the valuable suggestions and criticism. We've conducted further study and updated our draft. Due to the capacity limit, we placed the results in the Appendix, see Appendix A.4.
>
> Re Weaknesses:
>
> 1. We definitely agree that high-quality negative samples may promote the performance of discriminators. However, it is also not the case that ``negative samples'' almost as high-quality as the original golden sample would still promte the performance of the discriminator. In other words, the goal is that when the synthesized samples could still be considered negative, we want to maximize our chance of fooling weak discriminators (in other words, being close to the original distribution). One of the purpose of this paper is to focus how could we properly formulate such a goal (which essentially a min-max game) such that we can steadily expect a more efficient negative sampling process, following some pricipled practice. In this paper, our finding is to simply using CDM and enlarging the discrepancy of the amateur and expert model.
>
> 2. Thank you for your valuable feedback. We've included results from a few more recently published models in the updated draft. G-Eval is definitely the SOTA among all the studied metrics, yet we argue this could be a direct result that GPT-4 is explicitly trained to align with human opinions, making it less of a fair comparison. While T5 is rather an older model family, it is still very competitive (especially the larger configurations of it) in many import aspects. Detailed comparison and discussion can be found in the Appendix.
>
> 3. We apologize if the current writing confuses the reviewer. We struggle to fit all the discussions into the capacity limit. We've attached extended discussion and analysis in the appendix.

---

### Meta-Review · Area_Chair_vsEK · 2023-12-12

**Metareview:**

**Paper Summary:**

This paper introduces Contrastive Distribution Methods (CDM), an approach for evaluating open-domain text generation. It operates under the assumption that language models (LMs) improve as they increase in size. The authors propose two evaluation metrics: Generative CDM, which contrasts two LM distributions to generate negative examples for training discriminator-based metrics, and Discriminative CDM, which directly scores outputs based on differences in distributions. The paper presents experiments on multi-turn dialogue and factuality in abstractive summarization, aiming to demonstrate that CDM correlates more closely with human judgment than existing automatic evaluation metrics.

**Strengths:**

1. Novelty: The paper's application of contrastive momentum for evaluation is innovative (qwhh, V2uz, kEEE).
2. Promising Results: CDM outperforms several baselines in dialogue and summarization evaluation tasks (CKwf, V2uz, kEEE, 6ryG).

**Weaknesses:**

1. Applicability and Scalability: The method assumes contrastive momentum to derive a distribution that is presumably superior to the largest model in the family. This may not always be true, particularly as models increase in size and potentially yield diminishing returns.
2. Reliance on Partial Order Assumption: The methodology is based on the assumption that larger models inherently perform better, an assumption that may not always hold true (kEEE, V2uz, 6ryG).
3. Negative Sample Generation: The necessity of the approach for generating negative samples, as compared to a simpler method of direct sampling from the expert model, is questioned (CKwf, qwhh). The authors' rebuttal shows that larger experts performed worse, which contradicts the intuitive basis of the contrastive approach aiming to approximate the "Imaginary Oracle" depicted in Figure 2.
4. Comparison to Baselines: G-EVAL shows superior performance according to the authors' response to CKwf, which weakens the argument for CDM's superiority. However, I agree with the authors that this may not be a fair comparison due to GPT-4 being a proprietary model.

Additionally, reviewers have noted the absence of ablation studies, but it appears that many concerns are addressed in Table 7 in the appendix. It is recommended that the authors move this table to the main paper since ablation studies and comparisons to baselines are crucial.

**Decision:**

Considering the reviews, while the paper introduces an innovative approach with some promising results, there are concerns regarding the method's design and its generalizability. Therefore, I recommend against accepting this paper.

**Justification For Why Not Higher Score:**

Please refer to Weaknesses stated above.

**Justification For Why Not Lower Score:**

N/A

---

### Decision · Program_Chairs · 2024-01-16

Reject